# An Updated Greek National Checklist of Chondrichthyans

Ioannis Giovos [1,2,†], Roxani Naasan Aga-Spyridopoulou [1,†], Fabrizio Serena [3,†], Alen Soldo [4,†], Adi Barash [5], Nikolaos Doumpas [1], Georgios A. Gkafas [6], Dimitra Katsada [1], George Katselis [2], Periklis Kleitou [7], Vasileios Minasidis [1,2], Yannis P. Papastamatiou [8], Eleana Touloupaki [9] and Dimitrios K. Moutopoulos [2,*,†]

1   iSea, Environmental Organisation for the Preservation of the Aquatic Ecosystems, 54645 Thessaloniki, Greece
2   Department of Animal Production, Fisheries & Aquaculture, University of Patras, 30200 Mesolongi, Greece
3   Institute for Marine Biological Resources and Biotechnology, National Research Council (CNR-IRBIM), I-91026 Mazara del Vallo, Italy
4   Department of Marine Studies, University of Split, 21000 Split, Croatia
5   Sharks in Israel, NGO for the Conservation of Sharks and Rays, Amirim 1214000, Israel
6   Department of Ichthyology and Aquatic Environment, School of Agricultural Sciences, University of Thessaly, 38446 Volos, Greece
7   Marine and Environmental Research (MER) Lab, Limassol 4533, Cyprus
8   Institute of the Environment, Department of Biological Sciences, Florida International University, North Miami, FL 33181, USA
9   MEDASSET-Mediterranean Association to Save the Sea Turtles, 10672 Athens, Greece
*   Correspondence: dmoutopo@upatras.gr
†   These authors contributed equally to this work.

**Abstract:** Accurate checklists of species are essential for evaluating their conservation status and for understanding more about their distribution, biology and ecology and, therefore, the first step in order to effectively protect them. According to the existing literature, the Greek seas are rich in chondrichthyan biodiversity and herein, we update the most recent chondrichthyan checklist for the country regarding the species that are present in the Greek waters, correct unvalidated miscellaneous sightings and observations and provide guidelines about future research to improve their conservation. In total, 330 sources were collected from which 276 were used for further analysis, resulting in 1485 records of 67 species, among which 61 are confirmed by experts, including 34 sharks, 26 batoids and one chimaera. We are further listing six species as "Questionable/Not Confirmed".

**Keywords:** elasmobranchs; batoid; shark; chimaera; Eastern Mediterranean; Aegean Sea; Ionian Sea; Crete

## 1. Introduction

Chondrichthyans are a taxonomic group that contains about 1296 species worldwide [1]. The chondrichthyans belong to two subclasses, Holocephali (Chimaeras) and Elasmobranchii (sharks and batoids). These organisms have been living on earth for about 400 million years. In their majority, they are meso- to upper-level predators in marine ecosystems and may play an important ecosystem role. The Mediterranean Sea, despite its small acreage, is an important area for chondrichthyans, as it is characterized by moderate biodiversity [1], with 88 species already recorded at present [2]. Unfortunately, at the same time, it is probably the most impacted area for chondrichthyans, as they are the most threatened fish group. Particularly, from the 73 species populations in the Mediterranean Sea, which are assessed by the Red List of Threatened Species of the International Union for Conservation of Nature (IUCN), a percentage of 53.4% has been classified as Threatened and more than one-third as Data Deficient or Not Evaluated due to a lack of relevant data [3].

According to the existing literature, the Greek seas are rich in chondrichthyan biodiversity, with 36 species of sharks, 30 species of batoids and 1 species of chimaera being reported [4]. The Greek Red Book, published in 2009, includes 63 species of chondrichthyans, of which 50 are listed as "Not Assessed", while all the rest belong to one of the threatened categories [5]. In recent years, considerable efforts have been made to study the ecology and biology of these species, both in Greece but also in the Mediterranean [2]. However, knowledge of chondrichthyans in the Greek territorial waters is still limited, especially when it comes to their interaction with fisheries.

Accurate checklists of species are essential for evaluating their conservation status and for understanding more about their distribution and their biology and ecology [6–8]. This can help us understand more about the human impact on their populations and, consequently, design proper management action, supporting their conservation.

For this reason, a comprehensive review of the existing knowledge regarding chondrichthyan species' presence in the Greek territorial waters was conducted through a systematic and in-depth review of the current situation. The primary goal was to update the most recent chondrichthyan checklist for the country regarding the species that are present in Greek waters, correct possible miscellaneous sightings and observations and provide guidelines for future research in an attempt to increase their conservation.

## 2. Materials and Methods

A systematic literature review (up to December 2021) was conducted applying the Preferred Reporting Items for Systematic Reviews and Meta-Analyses approach [9]. Chondrichthyan records were collected from peer-reviewed publications archived in Google Scholar using the keywords "Greece" and "Greek" plus the search terms "chondrichthyan(s)", "chondrichthyes", "cartilaginous fish(es)", "elasmobranchii", "elasmobranch(s)", "shark(s)", "batoid(s)", "ray(s)", "skate(s)", "chimaera", "holocephali", "holocephalans" and "rabbitfish" to identify items with relevant titles, keywords or abstracts. We selected "anytime" for the publication date.

Further chondrichthyan records were searched through:

- **Government reports and policy documents**: In particular, the GR EU Data Collection Framework (DCF) reports, published between 2005 and 2019 and available at the following webpage: https://datacollection.jrc.ec.europa.eu/ars (accessed on 1 June 2021).
- **Grey literature**: This was explored through the online database HEAL-Link (Hellenic Academic Libraries Link; https://www.heal-link.gr/en/home-2/) (accessed on 21 December 2021).
- **Global Biodiversity Information Facility (GBIF)**: The Global Biodiversity Information Facility (GBIF) is the largest open access primary biodiversity database and contains over 1.5 billion species occurrence records.
- **Ocean Biodiversity Information System (OBIS)**: The Ocean Biodiversity Information System (OBIS), a global open access database on marine biodiversity for science, conservation and sustainable development, is focused on marine species and contains more than 6.5 million records for 137,215 species.
- **The Mediterranean Elasmobranchs Citizen Observations (MECO) Project**: The MECO project was launched in 2014 in response to enthusiastic scuba divers uploading pictures of sharks and rays from their dives. It aims to collate knowledge on chondrichthyan occurrence, seasonality and distribution using citizen science and social media. The project involves the collaboration of local scientists, which gradually expanded the operation to eleven countries and ten Facebook groups www.facebook.com/pg/theMECOproject (accessed on 31 December 2021). In MECO, participants report their sightings with photographic evidence. Scientific experts request further information when needed, such as date, location, specimen length and weight, number of individuals observed and depth of the observation (if applicable). The experts then check pictures for authenticity by using a Google automatic image

recognition tool and identify all original pictures to the lowest possible taxonomic level. Whenever possible, specimens are collected and experts record data such as maturity, gestation and sex. Finally, there is also a two-way dialogue between citizen participants and scientific experts to retrieve historical records based on old pictures and social media posts.

- **ByElasmoCatch**: The ByElasmoCatch project was launched in 2019 by iSea to assess the impact of fisheries on elasmobranchs in the North Aegean and collect information on species biology and ecology. Observations, measurements and samples are collected during monthly visits to fishing vessels. The project is ongoing (2022) and is funded by the Ocean Care and the Shark Foundation/Hai Stiftung.

- **MEDLEM**: The main aims of the MEDLEM program are (i) collecting information on bycatch, sighting and stranding events throughout the Mediterranean and Black seas, following a common protocol and (ii) recording their spatial occurrence. As an additional goal, MEDLEM stores scientific papers related to elasmobranchs as well as any reliable information from newspapers and social media. The MEDLEM program directly links up with the FAO IPOA-Sharks and has been endorsed by the SAC Sub-Committee on Marine Environment and Ecosystems (SCMEE) of the GFCM, Scientific Advisory Committee on Fisheries (SAC).

- **iNaturalist:** is a social network of naturalists, citizen scientists and biologists built on the concept of mapping and sharing observations of biodiversity across the globe. It is a joint initiative of the California Academy of Sciences and the National Geographic Society and currently numbers 100,000,000 verifiable observations.

An expert opinion was also used from researchers invited to participate in the preparation of the list. All researchers were asked to provide additional publications that contained original data regarding the presence of chondrichthyans in Greece that were not included in the database after the search in the abovementioned sources.

All sources (publications, reports, biodiversity databases) were organized in a single database including information regarding the species, the year of the sighting, the location (organized by Geographical subareas (GSAs) of General Fisheries Commission for the Mediterranean; Resolution GFCM/33/2009/2), the year of publication and a link to the publication. When a source included reports of observations from one GSA only one record was inserted in the database, while for sources that included reports of observations from different GSAs, we included as many records as the GSAs. Species names and families followed Eschmeyer's Catalog of Fishes [1] and the Red List of Threatened Species of the International Union for Conservation of Nature (IUCN).

*Exclusion and Inclusion Criteria*

Studies were considered vague when:

1. They did not provide sufficient information about the location of the observation and pictures of the individual(s) observed;
2. (For review papers) the original sources could not be tracked or the original sources do not provide sufficient information (location of the observation and pictures of the individual(s);
3. They were performed in fish and auction markets;
4. They were referring to ancient specimen remaining;
5. They were referring to museum collection specimens, of which the capture location was not provided.

Presence index in the Greek waters was based on the following criteria:

**Rare**: Few records over a longer period of time (decades);
**Occasional**: Recorded every few years;
**Common**: Few records recorded on a yearly basis;
**Abundant**: Often recorded in catches (or seen) on a yearly basis;
**Questionable/Not Confirmed**: Record needs confirmation.

Studies from Turkey referring to the GSAs that Greece and Turkey share were included due to the migratory nature of most studied species and the proximity between the two countries that share sea borders. Species records that included exclusively vague sources were considered as "Questionable/Not Confirmed" for Greek waters.

All studies before 1990 were excluded from further analysis, given that the aim of this work is to present an updated list of chondrichthyan species that still exist in the Greek waters.

## 3. Results

Source collection resulted in a total of 330 sources, from which 276 were used for further analysis while the rest were excluded (see Supplementary Materials). The analysis of the sources after 1990 resulted in 1485 records of 67 species, among which 61 are confirmed by expert, including 34 sharks, 26 batoids and one chimaera and belonging to 10 orders. Overall, 30 species of batoids, 1 chimaera species and 36 species of sharks were found through the review (Table 1). Based on the qualitative analysis of the sources and the expert's opinion, the Greek waters host 61 chondrichthyan species, represented by 26 batoids, 34 sharks and 1 chimaera (Figure 1). The six non-retained species are considered misidentification or questionable species.

**Table 1.** Number of sources presenting observations of the chondrichthyan species in Greece in available sources since 1990. No is the number of sources. Definitions are presented in Section 2.

| Order | Family | Species | Author | No | Status |
|---|---|---|---|---|---|
| | | **SELACHII** | | | |
| **HEXANCHIFORMES** | | | | | |
| | **Hexanchidae** | | Gray 1851 | | |
| | | *Heptranchias perlo* | (Bonnaterre, 1788) | 24 | Common |
| | | *Hexanchus griseus* | (Bonnaterre, 1788) | 29 | Common |
| | | *Hexanchus nakamurai* | Teng, 1962 | 4 | Rare |
| **LAMNIFORMES** | | | | | |
| | **Carchariidae** | | Müller & Henle, 1838 | | |
| | | *Carcharias taurus* | Rafinesque, 1810 | 10 | Rare |
| | **Odontaspididae** | | Müller & Henle, 1839 | | |
| | | *Odontaspis ferox* | (Risso, 1810) | 15 | Rare |
| | **Alopiidae** | | Bonaparte, 1835 | | |
| | | *Alopias superciliosus* | (Lowe, 1841) | 27 | Rare |
| | | *Alopias vulpinus* | (Bonnaterre, 1788) | 24 | Common |
| | **Cetorhinidae** | | Gill, 1861 | | |
| | | *Cetorhinus maximus* | (Gunnerus, 1765) | 18 | Rare |
| | **Lamnidae** | | Bonaparte, 1835 | | |
| | | *Carcharodon carcharias* | (Linnaeus, 1758) | 29 | Rare |
| | | *Isurus oxyrinchus* | Rafinesque, 1810 | 30 | Occasional |
| | | *Lamna nasus* | (Bonnaterre, 1788) | 14 | Rare |
| **CARCHARHINIFORMES** | | | | | |
| | **Scyliorhinidae** | | Gill, 1862 | | |
| | | *Scyliorhinus canicular* [§] | (Linnaeus, 1758) | 96 | Abundant |
| | | *Scyliorhinus stellaris* | (Linnaeus, 1758) | 20 | Common |
| | **Pentanchidae** | | Smith, 1912 | | |
| | | *Galeus melastomus* | Rafinesque, 1810 | 74 | Abundant |
| | **Triakidae** | | Gray, 1851 | | |
| | | *Mustelus asterias* | Cloquet, 1819 | 19 | Rare |
| | | *Mustelus mustelus* | (Linnaeus, 1758) | 47 | Common |
| | | *Mustelus punctulatus* | Risso, 1827 | 8 | Occasional |
| | | *Galeorhinus galeus* | (Linnaeus, 1758) | 25 | Rare |

**Table 1.** *Cont.*

| Order | Family | Species | Author | No | Status |
|---|---|---|---|---|---|
| | **Carcharhinidae** | | Jordan & Evermann, 1896 | | |
| | | *Carcharhinus brevipinna* | (Valenciennes, 1839) | 7 | Rare |
| | | *Carcharhinus obscurus* | (Lesueur, 1818) | 2 | Questionable/ Not Confirmed: |
| | | *Carcharhinus plumbeus* | (Nardo, 1827) | 19 | Occasional |
| | | *Prionace glauca* | (Linnaeus, 1758) | 47 | Common |
| | | *Rhizoprionodon acutus* | (Rüppell, 1837) | 3 | Rare |
| | **Sphyrnidae** | | Bonaparte, 1840 | | |
| | | *Sphyrna zygaena* | (Linnaeus, 1758) | 15 | Rare |
| **SQUALIFORMES** | | | | | |
| | **Dalatiidae** | | Gray, 1851 | | |
| | | *Dalatias licha* | (Bonnaterre, 1788) | 34 | Occasional |
| | **Etmopteridae** | | Fowler, 1934 | | |
| | | *Etmopterus spinax* | (Linnaeus, 1758) | 50 | Common |
| | **Somniosidae** | | Jordan, 1888 | | |
| | | *Centroscymnus coelolepis* | Barbosa du Bocage & de Brito Capello, 1864 | 6 | Questionable/ Not Confirmed: |
| | | *Somniosus rostratus* | (Risso, 1827) | 5 | Rare |
| | **Oxynotidae** | | Gill, 1863 | | |
| | | *Oxynotus centrina* | (Linnaeus, 1758) | 46 | Common |
| | **Centrophoridae** | | Bleeker, 1859 | | |
| | | *Centrophorus uyato* | (Rafinesque, 1810) | 35 | Common |
| | **Squalidae** | | de Blainville, 1816 | | |
| | | *Squalus acanthias* | Linnaeus, 1758 | 49 | Common |
| | | *Squalus blainville* | (Risso, 1827) | 64 | Abundant |
| **ECHINORHINIFORMES** | | | | | |
| | **Echinorhinidae** | | Gill, 1862 | | |
| | | *Echinorhinus brucus* | (Bonnaterre, 1788) | 8 | Rare |
| **SQUATINIFORMES** | | | | | |
| | **Squatinidae** | | de Blainville, 1816 | | |
| | | *Squatina aculeata* | Cuvier, 1829 | 22 | Rare |
| | | *Squatina oculata* | Bonaparte, 1840 | 19 | Rare |
| | | *Squatina squatina* | (Linnaeus, 1758) | 22 | Rare |
| | **BATOIDEA** | | | | |
| **TORPEDINIFORMES** | | | | | |
| | **Torpedinidae** | | Henle, 1834 | | |
| | | *Tetronarce nobiliana* | (Bonaparte, 1835) | 23 | Occasional |
| | | *Torpedo marmorata* | Risso, 1810 | 56 | Abundant |
| | | *Torpedo torpedo* | (Linnaeus, 1758) | 15 | Common |
| **RHINOPRISTIFORMES** | | | | | |
| | **Rhinobatidae** | | Bonaparte, 1835 | | |
| | | *Rhinobatos rhinobatos* | (Linnaeus, 1758) | 7 | Rare |
| | **Glaucostegidae** | | Last, Séret & Naylor, 2016 | | |
| | | *Glaucostegus cemiculus* | (Geoffroy St. Hilaire, 1817) | 8 | Rare |
| **RAJIFORMES** | | | | | |
| | **Rajidae** | | de Blainville, 1816 | | |
| | | *Dipturus* cf. *batis* * | (Linnaeus, 1758) | 6 | Questionable/ Not Confirmed: |
| | | *Dipturus oxyrinchus* | (Linnaeus, 1758) | 39 | Abundant |
| | | *Leucoraja circularis* | (Couch, 1838) | 13 | Occasional |
| | | *Leucoraja fullonica* | (Linnaeus, 1758) | 8 | Questionable/ Not Confirmed: |
| | | *Leucoraja melitensis* | (Clark, 1926) | 9 | Rare |
| | | *Leucoraja naevus* | (Müller & Henle, 1841) | 27 | Rare |
| | | *Raja asterias* | Delaroche, 1809 | 33 | Abundant |

**Table 1.** *Cont.*

| Order | Family | Species | Author | No | Status |
|---|---|---|---|---|---|
| | | *Raja brachyura* | Lafont, 1873 | 12 | Occasional |
| | | *Raja clavata* | Linnaeus, 1758 | 77 | Abundant |
| | | *Raja miraletus* | Linnaeus, 1758 | 47 | Abundant |
| | | *Raja montagui* | Fowler, 1910 | 25 | Rare |
| | | *Raja polystigma* | Regan, 1923 | 21 | Abundant |
| | | *Raja radula* | Delaroche, 1809 | 41 | Abundant |
| | | *Raja undulata* | Lacepède, 1802 | 10 | Rare |
| | | *Rostroraja alba* | (Lacepède, 1803) | 16 | Rare |
| **MYLIOBATIFORMES** | | | | | |
| | **Dasyatidae** | | Jordan & Gilbert, 1879 | | |
| | | *Bathytoshia lata* | (Garman, 1880) | 5 | Rare |
| | | *Dasyatis marmorata* | (Steindachner, 1892) | 1 | Rare |
| | | *Dasyatis pastinaca* | (Linnaeus, 1758) | 50 | Abundant |
| | | *Dasyatis tortonesei* | Capapé, 1975 | 5 | Questionable/ Not Confirmed: |
| | | *Pteroplatytrygon violacea* | (Bonaparte, 1832) | 8 | Abundant |
| | **Gymnuridae** | | Fowler, 1934 | | |
| | | *Gymnura altavela* | (Linnaeus, 1758) | 10 | Common |
| | **Aetobatidae** | | Agassiz, 1858 | | |
| | | *Aetomylaeus bovinus* | (Geoffroy St. Hilaire, 1817) | 10 | Occasional |
| | **Myliobatidae** | | Bonaparte, 1835 | | |
| | | *Myliobatis aquila* | (Linnaeus, 1758) | 26 | Occasional |
| | **Rhinopteridae** | | Jordan & Evermann, 1896 | | |
| | | *Rhinoptera marginata* | (Geoffroy St. Hilaire, 1817) | 2 | Questionable/ Not Confirmed: |
| | **Mobulidae** | | Gill, 1893 | | |
| | | *Mobula mobular* | (Bonnaterre, 1788) | 9 | Rare |
| | **CHIMAERAS** | | | | |
| **CHIMAERIFORMES** | | | | | |
| | **Chimaeridae** | | Rafinesque, 1815 | | |
| | | *Chimaera monstrosa* | Linnaeus, 1758 | 26 | Occasional |

§ Based on the recent study of [10], after the examination of *Scyliorhinus canicula* (Linnaeus, 1758) specimens from the Mediterranean and elsewhere the species was separated into two distinct species *S. canicula* and *Scyliorhinus duhamelii* with the examined specimens of the latter distributed along Adriatic and Mediterranean Seas, along the continental shelves of Croatia, Greece, Tunisia and Argelia. Therefore, it is possible some of the sources presented here refer to *S. duhamelii*; however, further research is required to confirm the presence of the species in Greece. * *Dipturus* cf. *batis* could refer either to *Dipturus batis*, or *Dipturus intermedius* or a mix of the two species. More research is required on these species.

Four batoids were listed as "Questionable/Not Confirmed" (*Dasyatis tortonesei*, *Dipturus* cf. *batis*, *Leucoraja fullonica* and *Rhinoptera marginata*), as well as two species of sharks (*Centroscymnus coelolepis* and *Carcharhinus obscurus*), while one shark species was considered as "Not valid" (*Sphyrna tudes*). For *Dipturus* cf. *batis*, the rationale based on [2] was followed. Although [11] stated that *D. nidarosiensis* might be involved, as it has been found in the Mediterranean, ref. [2] considers the species questionable for the Mediterranean and suspects that *Dipturus* cf. *batis* records refer to a species complex, including potentially *D. nidarosiensis* and *D. intemedius*. Regarding *C. coelolepis*, *C. obscurus* and *R. marginata*, the available sources did not provide sufficient evidence about the presence of the species in Greek waters, because either identification was vague through the photographic evidence presented or they were only mentioned in review papers, with original publications not being able to be tracked. *C. coelolepis* appeared in five records (Table 1) [4,12–15]. However, all publications, apart from [12,13], do not present original data but were referring primarily to the record of [12]. In [13], the authors claim that it "was believed to be either *Centrophorus granulosus* or *C. coelolepis*" and, thus, no definite conclusion can be made. In [12], *C. coelolepis* is listed among the species observed; however, no further information is provided nor a picture and, therefore, we considered the status of the species "Questionable/Not Confirmed" and further research is required.

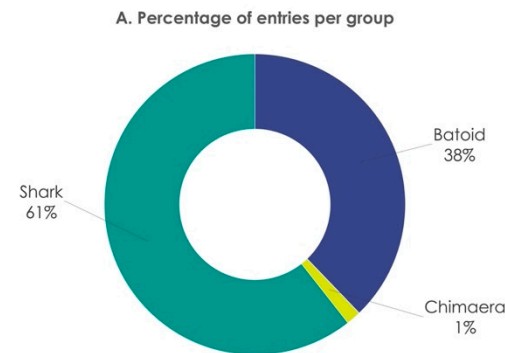

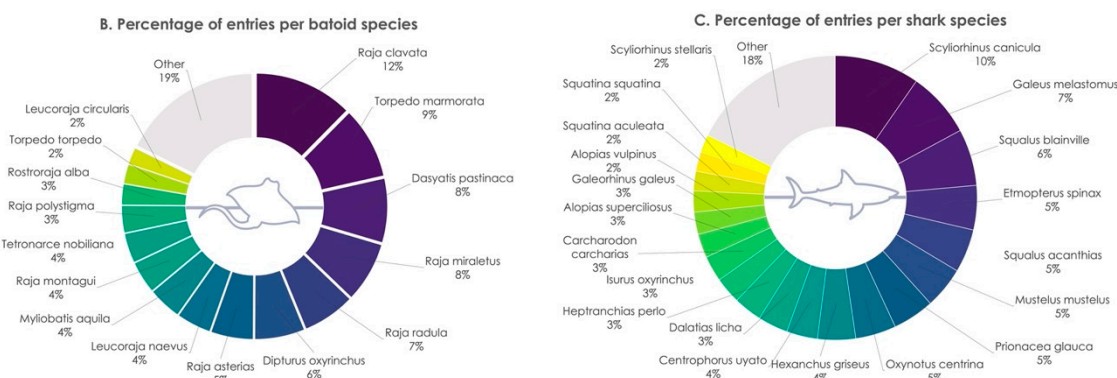

**Figure 1.** (**A**). Percentage of entries per group of chondrichthyans in Greek waters (**B**). Percentage of entries examined per species for batoids (**C**). Percentage of records per species for sharks. Species with less than 2% of records are not shown in the graph but can be found in Table 1.

In total, 1012 records were found for the Aegean Sea (GSA 22), 336 for the Ionian Sea (GSA 20) and 137 for Crete (GSA 23) (Figure 2). The majority of the records referred to sharks (60.31%), followed to a lesser extent by batoids (38.03%) and to a minor extent by chimaeras (1.66%), with the percentages slightly differing among the three GSAs (Figure 2).

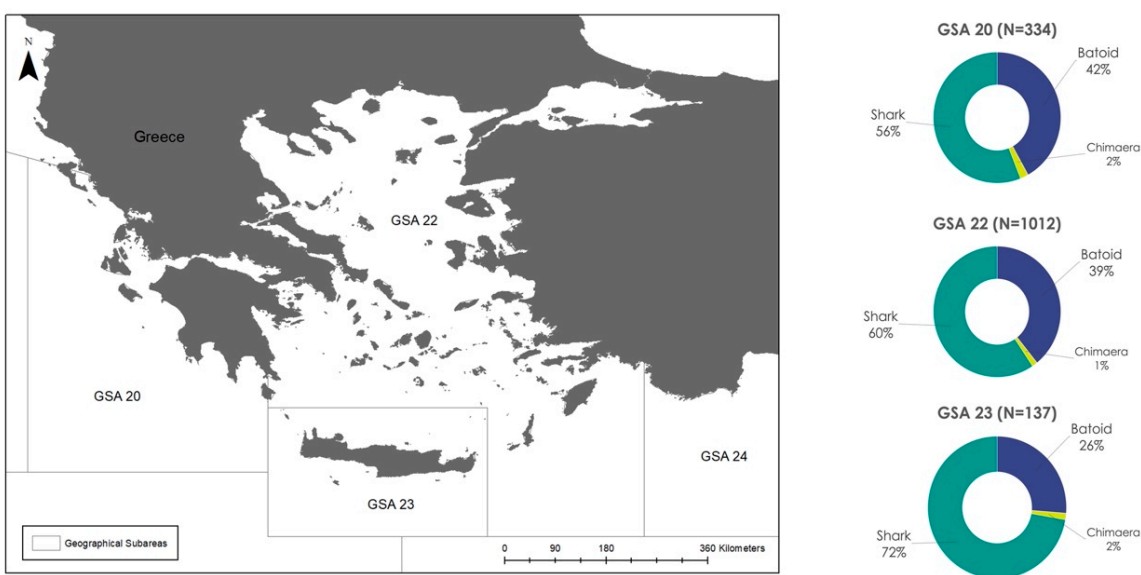

**Figure 2.** Map of the Geographic subareas (GSAs) in Greek waters and percentage of entries found that referred to sharks, batoids or chimaeras in the GSAs.

Two records of *Carcharhinus obscurus* appeared in our review (Table 1). The first was reported in the citizen-science platform iNaturalist and the second was reported by [16]. In both cases, we considered the photographic evidence not robust enough to confirm the presence of the species in Greek waters, while the records presented in [16] are dated back to 1942. In this work, we consider the current presence of the species in Greek waters as "Questionable/Not Confirmed" and we suspect that it might be a vagrant species.

*Rhinoptera marginata* appeared in three records (Table 1) [17,18], with all of them being review sources and not referring to any original observation of the species but to older reviews, such as [19–21]. Therefore, we considered the species presence in Greek waters as "Questionable/Not Confirmed". Regarding *D. tortonesei*, *D.* cf. *batis* and *L. fullonica*, we are following the suggestions by [2] that, for all these species, further investigation is required to confirm their presence in the Mediterranean and, thus, in Greece.

The vast majority of the batoid species (27 species out of 30) present in the Greek waters have not been assessed in the Greek Red Book [5] (Figure 3); however, from those, the Mediterranean population of 11 was assessed as threatened (Critically Endangered; Endangered; Vulnerable) in the IUCN Red List for Threatened Species. The same is true for sharks, with 25 of the 36 species not being assessed in the Greek Red Book (Figure 3), while the Mediterranean population of 12 (50%) was assessed as threatened (Critically Endangered; Endangered; Vulnerable) in the IUCN Red List for Threatened Species (Figure 3).

**Figure 3.** (**A**,**B**). Conservation status of the batoid and shark species, respectively, present in Greek waters, assessed in the Greek Red Book. (**C**,**D**). IUCN Conservation status of the Mediterranean population of the batoid and shark species, respectively, present but not evaluated in the Greek waters.

Regarding the research effort on elasmobranchs within Greek waters, it appears to be increasing rapidly in the last 10 years, having doubled compared to the period 2000–2020 and more than tripled from the period 1978–2000 (Figure 4). The increase in the research effort coincides with programs to evaluate fish stocks in European seas. In particular, the MEDITS program of the EU made it possible to produce numerous scientific papers, not only relating to the stock assessment but also concerning taxonomic items. This allowed one to update the faunal lists of many marine areas, including Greek seas [22–32].

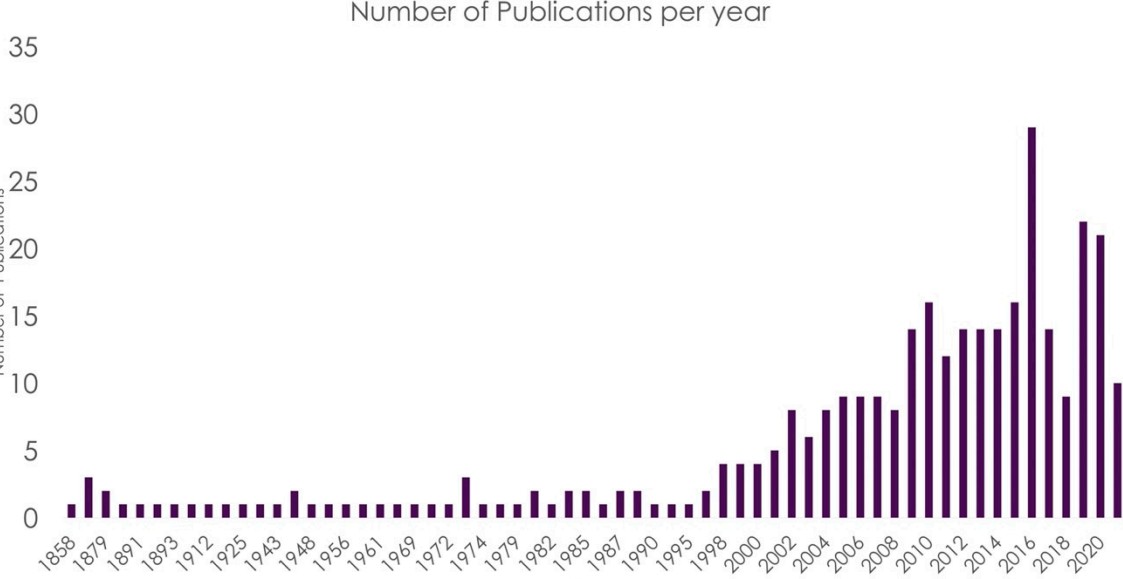

**Figure 4.** Number of scientific publications about elasmobranchs in Greek waters per year.

## 4. Discussion

Our work presents an exhaustive review of the available bibliography regarding occurrences of elasmobranchs in Greek waters, over almost two centuries (1858–2021). Given the analysis and the validation of the sources, we confirm the presence of 26 batoid species, 34 shark species and 1 chimaera, while we consider *D. tortonesei*, *D.* cf. *batis*, *L. fullonica*, *R. marginata*, *C. obscurus* and *C. coelolepis* as "Questionable/Not Confirmed" and further research is required for their confirmation. Potentially, some of these species are transient in Greek waters, while for others, further research is required regarding their taxonomical status in the Mediterranean Sea [2]. For the species listed as "Questionable/Not Confirmed", a dedicated campaign on social media, local mass media and peer-to-peer with fishers is required to confirm their presence in Greek waters. The latter might be beneficial for some other species, currently listed as "Rare", such as *Echinorhinus brucus* and *Rhinobatos rhinobatos*, for which the last observation was reported decades ago. Further research is also needed for some others (such as *S. canicula*) in order to validate the most recent findings of *S. duhamelii* revalidation from Greece.

The difference in the sources and the records among GSA 20, 22 and 23 cannot be attributed to differences in the abundance of the elasmobranch species (the study did not assess by the abundance of elasmobranchs in Greek waters), but we believe that it is more related to the scientific effort that seems to be very limited around Crete. Moreover, a number of records from the Ionian Sea are, to a small extent, duplicate records, resulting from publications of a project that conducted field work and utilized multi-dimensional research on the same species. For example, in the context of the project CoralFISH (https://imbriw.hcmr.gr/coralfish/, accessed on 23 December 2021), several publications were published, including on the diet and feeding strategy of blackmouth catshark *Galeus melastomus* and on the reproductive biology and length–weight relationships of *G. melastomus* in the eastern Ionian Sea, with both studies using the same specimens

and, therefore, occurrences [30,31]. The same is true for some studies in the Aegean Sea; however, the large number of publications from this GSA undermines the issue. Following the aforementioned research, we suggest that future research in Greece could focus more on Crete and then in the Ionian Sea. Apart from the fact that possibly more species might be present in these two areas, such limited research might result in insufficient understanding about the role of elasmobranch species in these areas and the threats they face.

It is also important to highlight the knowledge gap in the evaluations of elasmobranch species in the Greek Red Book, which was published in 2009 [5]. From the 61 species definitely present in the Greek waters, 50 (≈82%) were not evaluated due to a lack of data. Since then (2009) [5] a few more publications focusing on Greek waters have been published. Hence, a Red Book update is on its way in Greece as an initiative of the Ministry of Environment and Energy. For this reason, it is of particular importance to fulfill the following:

(i)     Dedicated research to take place on measuring the impact of fisheries and other human activities on the populations of elasmobranchs in Greek waters;

(ii)    Research centers participating in the national Data Collection Framework to utilize all the collected information, particularly from the MEDITS survey for providing abundance estimates for species in different locations around Greece;

(iii)   Funders to support initiatives in the country that aim to estimate the impact of fisheries on the population of elasmobranchs or that provide population abundance estimates, especially in Crete and the Ionian Sea.

## 5. Conclusions

Our study update the most recent chondrichthyan checklist for Greek waters, correct unvalidated miscellaneous sightings and observations, and provide several avenues for further research in an effort to improve chondrichthyan conservation. Although the Greek seas are rich in chondrichthyan biodiversity, only during the last decade research effort on this class have been largely increased mostly attributed to MEDITS program. However, in several locations, there are still significant gaps in knowledge (e.g., Corinth Gulf; Crete, etc.) and a lack of understanding about the pressure that several métiers appear to have on them (e.g., in the North Aegean Sea; [32]). Before the Red Book is updated, it is critical to fill these gaps in order to produce a comprehensive assessment of the elasmobranch species found in Greek waters.

**Supplementary Materials:** The following supporting information can be downloaded at: https://www.mdpi.com/article/10.3390/fishes7040199/s1.

**Author Contributions:** Conceptualization: I.G.; Methodology: I.G., D.K.M., F.S., A.S. and R.N.A.-S.; Validation: F.S. and A.S.; Formal Analysis: I.G., R.N.A.-S. and D.K.M.; Data Collection: I.G., R.N.A.-S., A.B., N.D., G.A.G., D.K., G.K., D.K.M., P.K., V.M., Y.P.P. and E.T.; Data Curation: I.G. and R.N.A.-S.; Writing—Original Draft Preparation: I.G. and D.K.M.; Writing—Review and Editing: All co-authors; Visualization, I.G.; Supervision: D.K.M., F.S. and A.S.; Project Funding Acquisition: I.G. All authors have read and agreed to the published version of the manuscript.

**Funding:** This research was funded by the Green Fund of the Hellenic Ministry of Environment and Energy in the context of the project "Updating of the Greek National Chondrichthyans Checklist". Additionally, the Shark Conservation Fund, the Save Our Seas Foundation, OceanCare and Shark Foundation/Hai Stiftung have been supporting projects of iSea that contributed significant data for this work.

**Institutional Review Board Statement:** Not applicable.

**Data Availability Statement:** The dataset analyzed in the current study is available from the corresponding author upon reasonable request.

**Acknowledgments:** We would like to warmly thank all fishers and citizen scientists for contributing to this work.

**Conflicts of Interest:** The authors declare no conflict of interest.

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
