# Peer review of "An Updated Greek National Checklist of Chondrichthyans"

_fishes, doi:10.3390/fishes7040199_

Round 1

Reviewer 1 Report

-Lines 27-29: what about reference [4]? They apparently managed to assess most of the species and provided a checklist, which is updated in the present manuscript.

-Line 40: preferably, a peer-reviewed paper should be cited instead of an electronic reference, such as Weigmann (2016: Annotated checklist of the living sharks, batoids and chimaeras (Chondrichthyes) of the world, with a focus on biogeographical diversity. Journal of Fish Biology 88(3), 837–1037. https://doi.org/10.1111/jfb.12874).

-Lines 41-44: citations needed.

-Lines 44-46: 88 species does not indicate high biodiversity. See Weigmann (2016) for biodiversity hotspots of chondrichthyans and rephrase the sentence.

-Line 46-48: citation needed. Dulvy et al. (2014: Extinction risk and conservation of the world’s sharks and rays. eLife 2014;3:e00590. DOI: 10.7554/eLife.00590) identified three main hotspots where the biodiversity of sharks and rays was particularly seriously threatened - the Indo-Pacific Biodiversity Triangle, Red Sea, and the Mediterranean Sea - and argue that national and international action is needed to protect them from overfishing.

-Lines 48-52: check the IUCN Red List website if this data is still up to date.

-Lines 67-70: I do not agree that the present manuscript sets a base line as it is not the first checklist of chondrichthyans in Greek waters. Please rephrase.

-Lines 74-77: Important search terms are missing, such as other words describing the geographic area, e.g., Greek, and other terms referring to chondrichthyans, e.g., Chondrichthyes, cartilaginous fish(es), elasmobranchii, holocephali, holocephalans.

-Lines 130-132: references [1, 9] are both Eschmeyer’s Catalog of Fishes (only different subsections). Therefore, delete one of them and just cite the catalog in general. Instead, you can add another references, such as Weigmann (2016).

-Lines 132-143: correct typo “wasnot”.

-Lines 155-157: I do not agree with the exclusion of data published before 1990. Instead, you can include the data and add a remark that the species has not been found since 1990.

-Results: according to the supplementary material, museum specimens were excluded. I can understand that museum specimens without detailed location information are excluded, but museum specimens with precise collection information definitely constitute valuable data sources and should be included.

-Results: add a list of voucher specimens from Greek waters for all species, following the data published by Ehemann et al. (2019: Ehemann, NR, González-González, LDV, Tagliafico, A, Weigmann, S. Updated taxonomic list and conservation status of chondrichthyans from the exclusive economic zone of Venezuela, with first generic and specific records. J Fish Biol. 2019; 95: 753– 771. https://doi.org/10.1111/jfb.14061).

-Lines 151-152: I do not agree with the general comment about migratory nature of species. The species studied include species with strongly different habits and this sentence needs to be rephrased and a citation be added.

Table 1: Dipturus is problematic in the Mediterranean. I think that D. nidarosiensis is a better candidate species for Dipturus-skates different from D. oxyrinchus in the Mediterranean Sea, see, e.g., Carbonara et al. (2019: Carbonara P, Cannas R, Donnaloia M, Melis R, Porcu C, Spedicato MT, Zupa W, Follesa MC. On the presence of Dipturus nidarosiensis (Storm, 1881) in the Central Mediterranean area. PeerJ. 2019 May 27;7:e7009. doi: 10.7717/peerj.7009). According to Last et al. (2016: Last, P.R., Séret, B., Stehmann, M.F.W. & Weigmann, S. (2016) Skates, Family Rajidae. In: Last, P.R., White, W.T., Carvalho, M.R. de, Séret, B., Stehmann, M.F.W & Naylor, G.J.P (Eds.) Rays of the World. CSIRO Publishing, Melbourne, pp. 204–363), both Dipturus batis and D. intermedius were historically found in the Mediterranean Sea but are now extirpated from this area. Is the indication of D. cf. batis possible based on historical data? A discussion citing the aforementioned references is recommended.

-Lines 172 ff: have you tried to verify all the records mentioned, e.g., by examining photographs or asking the authors of the respective papers for more information and images? Examination of photographs is only mentioned for part of the species.

Author Response

In the attached file, we provide detailed responses to the comments, which were all also included in the revised MS.

Author Response

In the attached file we provide detailed responses to the comments, which were all alos included in the revised MS.

Reviewer 3 Report

Comments:  

Line 35 and 164: numbers under 10 must be written with letters unless a metric unit accompanies them

Line 155: How do you know that a specie that was reported before 1990 but does not appear in further publications doesn’t exist currently in the Greek waters? It could be a very rare species, for example in deep waters, that is not usually observed by divers or is not present in fisheries, and because of that, you can’t be sure if the species is still there or not. An update doesn´t mean only taking into account the more recent information.

Table 1: Some of the names of species, families, and orders were cut so the spelling is not correct for example Odontaspididae, Carcharhiniformes, Hexanchiformes.

Hexanchus nakamurai. Based on Ebert et al 2021 this specie is not present in the Atlantic Ocean and is commonly confused with H. griseus. So your report is the first report of this specie for the Atlantic Ocean? Do you confirm this report?

Sphyrna tudes is a valid specie based on the Echmeyer´s catalog of fishes you cited here

Organize the orders in this table from the most antique to the most recent. I recommend following the order for orders, families, and species proposed by Ebert et al 2021.

In this table there are 37 species of sharks, not 36 as you mention in line 162

Line 175. What is the evidence to consider S. tudes as a not valid specie? Since you mention in methods lines 129 to 131 that you followed Eschmeyer’s Catalog of Fishes and in this catalog they report as valid this specie.

Line 233. What is the meaning of etc. here in a list of references? Also, the format doesn’t follow here the citation format.

Line 228 and figure 4. Do you mention in methods that you were using only references after 1990 and here you area including references since 1858? So finally what do you do? Be consistent between methods and the results presented here

Line 264. Please do not repeat again the name of the specie

Line 234-292. Discussion

Do you mention in the first lines that you reviewed information from over the last century but in methods you mention you reviewed reports published after 1990, again this is not the same. Please clarify this either in the methods or in the rest of the manuscript.  

The data need to be discussed more deeply, authors need to show what the new information presented here compared with the past information of the Chondrichthyan’s biodiversity in Greece. What are the new reports? If the species presented here are the same species presented in other publications why is this work so relevant? The discussion needs to be rewritten by doing a deep analysis of the data.  

Author Response

In the attached file we provide detailed responses to the comments, which were all also included in the revised MS.

Round 2

Reviewer 1 Report

I thank the authors for their thorough revision. The manuscript can now be accepted for publication but I would still recommend adding a comment that D. nidarosiensis might be involved as it has been found in this general region following Last et al. (2016) as mentioned in my first review (and there is a typo in this sentence: "Morre").

Author Response

We are grateful for the positive feedback and detailed consideration. We have considered every comment and attempted to address each, as detailed in the attached letter.

Reviewer 3 Report

The comments about the discussion in my previous review weren´t addressed. The manuscript has valuable information but the authors didn´t discuss their findings. I expect a more profound discussion because they mention that there is previous information about the Chondrichthyan diversity in Greece in the introduction. Hence, if you have this information why not compare the numbers, find the differences and try to explain this difference? 

Table 1, I recommend using the same font size inside the table

Figure 4. The size of the figure is too big and doesn´t fit on the page of the manuscript 

Author Response

(The authors gave the same response as above.)
